# Coupling the Cardiac Voltage-Gated Sodium Channel to Channelrhodopsin-2 Generates Novel Optical Switches for Action Potential Studies

**DOI:** 10.3390/membranes12100907

**Published:** 2022-09-20

**Authors:** Christian vom Dahl, Christoph Emanuel Müller, Xhevat Berisha, Georg Nagel, Thomas Zimmer

**Affiliations:** 1Institute of Physiology II, University Hospital Jena, Friedrich Schiller University, 07740 Jena, Germany; 2Institute of Physiology—Neurophysiology, Julius Maximilians University, 97070 Wuerzburg, Germany

**Keywords:** optogenetics, channelrhodopsin, voltage-gated Na^+^ channel, action potential, delayed rectifier potassium channel, hERG, long QT syndrome

## Abstract

Voltage-gated sodium (Na^+^) channels respond to short membrane depolarization with conformational changes leading to pore opening, Na^+^ influx, and action potential (AP) upstroke. In the present study, we coupled channelrhodopsin-2 (ChR2), the key ion channel in optogenetics, directly to the cardiac voltage-gated Na^+^ channel (Na_v_1.5). Fusion constructs were expressed in *Xenopus laevis* oocytes, and electrophysiological recordings were performed by the two-microelectrode technique. Heteromeric channels retained both typical Na_v_1.5 kinetics and light-sensitive ChR2 properties. Switching to the current-clamp mode and applying short blue-light pulses resulted either in subthreshold depolarization or in a rapid change of membrane polarity typically seen in APs of excitable cells. To study the effect of individual K^+^ channels on the AP shape, we co-expressed either K_v_1.2 or hERG with one of the Na_v_1.5-ChR2 fusions. As expected, both delayed rectifier K^+^ channels shortened AP duration significantly. K_v_1.2 currents remarkably accelerated initial repolarization, whereas hERG channel activity efficiently restored the resting membrane potential. Finally, we investigated the effect of the LQT3 deletion mutant ΔKPQ on the AP shape and noticed an extremely prolonged AP duration that was directly correlated to the size of the non-inactivating Na^+^ current fraction. In conclusion, coupling of ChR2 to a voltage-gated Na^+^ channel generates optical switches that are useful for studying the effect of individual ion channels on the AP shape. Moreover, our novel optogenetic approach provides the potential for an application in pharmacology and optogenetic tissue-engineering.

## 1. Introduction

In most excitable cells, the fast upstroke of the action potential (AP) is caused by a temporal increase in the sodium (Na^+^) conductance through voltage-gated Na^+^ channels [1,2,3,4]. Naturally, these channels consist of two different types of subunits. The large pore-forming α subunit allows for the ion flux through the plasma membrane, while one or more of the four accessory β subunits modulate the α subunit function [5,6]. In mammals, nine structurally-related Na^+^ channel alpha subunits have been detected and functionally characterized (Na_v_1.1 to Na_v_1.9) [1,5,7]. They are often grouped either by their tissue-specific expression as neuronal, skeletal muscle, or cardiac Na^+^ channels or by their binding affinity to the neurotoxin Tetrodotoxin (TTX) as TTX-sensitive or -resistant. Na_v_1.1, Na_v_1.2, and Na_v_1.3 are mostly expressed in the central nervous system, Na_v_1.4 and Na_v_1.5 in skeletal muscle and cardiomyocytes, respectively. Na_v_1.7, Na_v_1.8, and Na_v_1.9 are preferentially localized in peripheral neurons, and Na_v_1.6 occurs in both the central and peripheral nervous system [1,3,5]. Most neuronal isoforms including Na_v_1.4 can be blocked by low nanomolar TTX concentrations (IC_50_ ~ 10 nM), whereas cardiac Na_v_1.5, as well as neuronal Na_v_1.8 and Na_v_1.9, are resistant to low TTX concentrations (IC_50_ ~1 to 100 µM) [5,8,9].

In human ventricular cardiomyocytes, AP upstroke and change in membrane polarity are mediated by Na_v_1.5 (phase 0) [5,8,10]. After short initial repolarization (phase 1), which is triggered by the transient outward potassium (K^+^) current *I*_to_ and facilitated by efficient Na_v_1.5 inactivation, increasing inward calcium (Ca^++^) and delayed outward K^+^ currents equalize each other, thereby forming the typical AP plateau (phase 2). Subsequently, rising K^+^ currents repolarize the membrane (phase 3) back to the resting potential (phase 4) [11,12]. For this repolarization, mainly two delayed rectifier K^+^ channels are essential: K_v_7.1, encoded by *KCNQ1* and generating *I*_Ks_ (slow *I*_K_), and K_v_11.1, encoded by *KCNH2* and generating *I*_Kr_ (rapid *I*_K_) [13,14]. The latter K^+^ channel is better known as hERG (human Ether-à-go-go-Related Gene). Loss-of-function mutations in both *KCNQ1* and *KCNH2* can cause a severe delay in AP repolarization. Clinically, such a dysfunction manifests as a prolongation of the QT interval in surface ECG recordings and is highly associated with life-threatening arrhythmic events. Such channelopathies are called long QT (LQT) syndrome, and depending on the defective gene locus, the disease can be further classified as LQT1 (K_v_7.1) or LQT2 (hERG) [15,16,17]. In contrast to loss-of-function mutations in delayed rectifier K^+^ channels, the third type of LQT syndrome (LQT3) is caused by de novo or inherited mutations in *SCN5A*, the gene encoding Na_v_1.5 [5,8,18,19,20,21]. In these cases, prolonged QT intervals result from an increased Na^+^ current during the AP (gain-of-function). In many cases, the additional Na^+^ influx results from defective Na^+^ channel inactivation [19]. The first and best-studied mutant channel is characterized by a three amino acid deletion in the inactivation gate (∆KPQ) [18]. In Na_v_1.5-∆KPQ mutant channels, the non-inactivating or persistent current fraction at the end of depolarizing voltage pulses can account for up to 5% of the initial peak current, a sustained conductance that is nearly 10-fold larger compared to corresponding measurements for wild-type Na_v_1.5 [18,22].

Channelrhodopsins represent another protein family that can change the membrane potential upon activation. In contrast to Na^+^, Ca^++^, and K^+^ channels shaping mammalian APs, they do not respond to voltage changes, but to light stimuli. The best-known member, channelrhodopsin-2 (ChR2), serves the green algae *Chlamydomonas reinhardtii* as a sensor for phototaxis [23,24]. ChR2 consists of seven transmembrane helices, forming the pore, and a retinal molecule, anchored via Schiff-base linkage to lysine 257 and forced to undergo isomerization upon blue-light illumination [25]. The resulting conformational changes finally lead to opening of the pore and influx of protons, and to a lesser extent, of other mono- and divalent cations [23,24,26]. Heterologous expression in *Xenopus laevis* oocytes and mammalian cells greatly facilitated our understanding of the gating mechanism leading to a four-state photocycle model with two conducting and two non-conducting states [26,27,28].

Taking advantage of the unique ChR2 properties as a light switch, several groups independently controlled AP excitation in neurons via blue light with millisecond resolution by expressing ChR2, thereby establishing the new field of optogenetics [29,30,31,32,33]. Despite the fascinating progress in recent years, there are still a few principal problems. For example, the single-channel conductance of ChR2, estimated to about 40–60 fS, is very low [24,34]. Consequently, ChR2 expression in target cells must be high enough either for the ChR2-mediated AP upstroke or for activating the surrounding voltage-gated Na^+^ channels. The intracellular targeting of ChR2 to cellular spots with a high Na^+^ channel density, such as the axon hillock or the intercalated disc region, however, cannot be easily controlled [35]. In an attempt to direct ChR2 in close vicinity to voltage-gated Na^+^ channels, we recently pursued two strategies [36]. First, we linked the optimized ChR2-T159C variant to the Na^+^ channel β1 subunit. This approach led to blue-light triggered APs in *Xenopus laevis* oocytes when co-expressing various Na^+^ channel α subunits. Second, we directly coupled ChR2 either to the Na_v_1.5 C- or N-terminus. This strategy, however, was unsuccessful in our previous study, which was mainly due to a complete loss of light-sensitive photocurrents in the heteromeric fusions [36].

In the present study, we introduce improved light-sensitive Na_v_1.5-ChR2 channels for AP generation. Using one of these novel optical switches, we show in *Xenopus laevis* oocytes the specific effect of two functionally distinct delayed rectifier K^+^ channels (K_v_1.2 and hERG), and the Na_v_1.5 deletion mutant ΔKPQ on AP repolarization. Our results suggest that the new optogenetic tools could be of importance to study the effect of various individual ion channels on the AP shape, for drug testing, and future optogenetic studies in naturally excitable cells.

## 2. Materials and Methods

### 2.1. Expression Plasmids and Recombinant DNA Procedures

For the expression of human Na_v_1.5 (*SCN5A*; accession number M77235; [8]), rat K_v_1.2 (*KCNA2*; accession number X16003; [37]), human K_v_11.1 (*hERG* or *KCNH2*; accession number U04270; [38]), and LQT3 deletion variant Na_v_1.5-ΔKPQ [18] we used plasmids pTSV40G-hNa_v_1.5 [39], pAKS2-RCK5 [37,40], pGEMHEnew-hERG (kindly provided by Thomas Baukrowitz, University of Kiel, Germany), and pTSV40-ΔKPQ [41], respectively. Plasmids coding for the five different Na_v_1.5-ChR2-ChR2 constructs (Na_v_1.5-CC-1 to Na_v_1.5-CC-5; Figure 1) and for the ChR2-ChR2-Na_v_1.5 fusion (CC-Na_v_1.5; Figure 1) were assembled by overlapping PCR using a thermostable DNA polymerase with proofreading activity (Pfu DNA polymerase, Promega, Madison/USA), and using suitable sites for restriction enzymes. As starting DNA templates, we used pEYFP-hH1 coding for Na_v_1.5-YFP [42], pGEM-β1-ChR2-YFP coding for β1-ChR2-YFP [36], pGEM-β1-ChR2-ChR2 coding for β1-ChR2-ChR2 [36], pTSV40G-hNa_v_1.5-ChR2 coding for Na_v_1.5-ChR2 [36], and pGEMHE-ChR2-T159C::YFP coding for ChR2-YFP [36,43,44]. To facilitate genetic engineering work, sequences for unique restriction sites were often incorporated at the 5′-end of forward and reverse PCR primers. All six coding regions were placed downstream to the T7 promoter either in pTSV40G [39] or pGEMHEnew [45]. The correctness of all plasmid constructs was confirmed by restriction analysis and, in the case of the PCR-derived sequences, by DNA sequencing (Seqlab Microsynth, Göttingen, Germany). All ChR2 sequences contained the T159C mutation to increase the binding affinity to endogenous retinal [43,44]. A detailed description of recombinant DNA procedures including the precise amino acid composition of the six artificial Na_v_1.5/ChR2 heteromers can be found in “Appendix A”.

### 2.2. Heterologous Expression in Xenopus leavis Oocytes

In vitro transcription was done using the mMESSAGE mMACHINE^®^ T7 Transcription Kit (Thermo Fisher Scientific, Waltham, MA, USA) after linearization of the pGEM- and pTSV40-/pTSV40G-derived plasmids with *Not*I, except for pTSV40G-hH1R (*Xba*I). In the case of pAKS2-RCK5, we used the mMESSAGE mMACHINE^®^ SP6 Transcription Kit (Thermo Fisher Scientific, Waltham, MA, USA), after linearization with *Eco*RI. Size and quality of cRNAs were first checked by agarose gel electrophoresis, and subsequently, the cRNA concentration was determined using a NanoDrop 2000c spectrophotometer (Thermo Fisher Scientific, Waltham, MA, USA). All preparations were adjusted to a standard concentration of 0.2 µg/µL and stored at −80 °C until use. *Xenopus laevis* oocytes were obtained from EcoCyte Bioscience (Dortmund, Germany). Before injection, cRNA preparations were diluted appropriately, so that the desired whole-cell current range could be obtained. The injection volume was 50 to 70 nl per oocyte. The amount of cRNA (ng/Oocyte) is indicated in legends to the respective Figures/Tables. Oocytes were incubated at 18 °C for 48 to 72 h in Barth medium (mM: 84 NaCl, 1 KCl, 2.4 NaHCO_3_, 0.82 MgSO_4_, 0.33 Ca(NO_3_)_2_, 0.41 CaCl_2_, 7.5 Tris/HCl, pH 7.4) containing 1 µM all-trans Retinal (Sigma). Experiments were repeated using two to seven additional batches of oocytes.

### 2.3. Electrophysiological Recordings

Recordings were performed at room temperature using a two-microelectrode setup and the TEC-05-S amplifier from npi Electronic Instruments (Tamm, Germany), essentially as already described [36,46]. Glass microelectrodes were filled with 3 M KCl, and microelectrode resistance was between 0.1 and 0.6 MΩ. The bath solution contained (in mM): 96 NaCl, 2 KCl, 1.8 CaCl_2_, 1 MgCl_2_, 10 HEPES/KOH, pH 7.4. Membrane resistance of oocytes was calculated from the injected current that was required to set the membrane voltage from the actual resting potential to −80 mV. All oocytes with a membrane resistance smaller than 100 kΩ were excluded from our measurements. Recording and data analysis were done on a PC with the ISO2 software (MFK, Niedernhausen, Germany). Sampling rate was usually 5 kHz, except for Na^+^ channel measurements (20 to 50 kHz).

Currents through Na^+^ channels (*I_Na_*) were elicited from a holding potential of −120 mV to test potentials ranging from −80 mV to 50 mV in 5 or 10 mV increments. If not otherwise stated, pulsing frequency was 1.0 Hz. Steady-state activation (m_∞_) was determined by fitting the Boltzmann equation m_∞_ = {1 + exp[−(V − V_m_)/s]} ^−1^ to the normalized conductance as function of voltage. Steady-state inactivation (h_∞_) was evaluated with a double-pulse protocol consisting of 500 ms prepulses to voltages between −140 and −30 mV followed by a constant test pulse of 10 ms duration to -20 mV (pulsing frequency: 0.5 Hz). The amplitude of peak *I_Na_* during the test pulse was normalized to the maximum peak current and plotted as a function of the prepulse potential. Data were fitted to the Boltzmann equation h_∞_ = {1 + exp[(V − V_h_)/s]} ^−1^. V_h_ and V_m_ are the mid-inactivation and mid-activation potentials, respectively. V and s are the respective test potentials and slope factors, respectively. Recovery from the inactivated state was determined with the following double-pulse protocol: 500 ms prepulse to −10 mV, followed by a variable recovery interval at −120 mV and a 20 ms test pulse to −20 mV. Pulsing frequency was 0.2 Hz to allow for full recovery of the Na_v_1.5 channel variants between the double pulses. Recovery curves were fitted using the following bi-exponential equation: *I* = 1 − [A_1_ × exp(−t/τ_1_) + A_2_ × exp(−t/τ_2_)]. *I* is the normalized current, t is the recovery time, and τ_1_ and τ_2_ are time constants of fast- and slow-recovering components, respectively.

The persistent Na^+^ current fraction in Na_v_1.5-ΔKPQ was determined as the sustained current at the end of a 140 ms test pulse to −15 mV that could be specifically blocked by 10 µM tetrodotoxin (TTX). Because the cardiac Na_v_1.5 channels are relatively resistant to TTX, we increased the sensitivity toward this toxin by reducing the external Na^+^ concentration, as previously described [47]. The respective bath solution contained (in mM): 20 NaCl, 78 KCl, 1.8 CaCl_2_, 1 MgCl_2_, 10 HEPES/KOH, pH 7.4. At the same time, we had to compensate for the reduced driving force and smaller Na^+^ currents by increasing the Na_v_1.5-ΔKPQ expression. Therefore, we injected undiluted cRNA (20 ng/oocyte), so that the whole-cell peak currents were in the range of 0.5 to 3.5 µA. Upon application of 10 µM TTX, at least 95% of the transient current amplitude could be blocked.

If not otherwise stated, whole-cell currents through voltage-gated K^+^ channels were elicited from a holding potential of −100 mV to test potentials ranging from −50 mV to 40 mV, followed by a third test pulse to 0 mV (K_v_1.2) or −50 mV (hERG). Pulsing frequency was between 0.2 and 0.4 Hz.

To elicit photocurrents, we attached a self-made blue LED device to the measurement chamber of the two-microelectrode setup (maximum at 472 nm). Light pulses were elicited by the ISO2 trigger function. Light intensity was measured by the LaserCheck power meter from Coherent (Dieburg, Germany), resulting in 7.1 mW/mm^2^. In voltage-clamp experiments, the holding potential was −80 mV, and the duration of the blue-light pulse was 1 sec, if not stated otherwise. The photocurrents generated by the six fusion channels were significantly smaller than those previously observed with β1-ChR and β1-ChR-YFP (compare the photocurrents in Table 1 and [36]). To increase the light-induced inward current and thus the likelihood to reach the threshold potential, we mounted a second LED to our setup in order to increase the blue-light exposed oocyte surface (measurements in Figures 3–9 and Appendix A). Voltage changes were recorded in the current-clamp mode, as follows: First, we injected an appropriate current to set the membrane potential from the intrinsic oocyte resting potential to −80 mV (usually between 50 nA and 150 nA). Second, a 50 ms blue-light pulse was applied to activate the tandem ChR2 variants in our fusion channels. Third, voltage changes were recorded for 1–4 s. Finally, measurements were repeated after holding the oocytes at −80 mV in our light-protected measurement chamber.

Data are presented as mean ± SEM. To test for statistical significance, an unpaired Student’s t-test was used. Statistical significance was assumed for *p* < 0.05.

## 3. Results

### 3.1. Functional Coupling of ChR2 to Cardiac Na_v_1.5

In order to obtain functional ChR2-Na_v_1.5 and Na_v_1.5-ChR2 fusion channels, we modified our previous constructs as follows: First, we coupled ChR2 as a dimer to Na_v_1.5, either at its C- or N-terminus. Second, we took into account that one of our previous β1-ChR constructs, containing an intracellularly exposed yellow fluorescent protein (YFP), a Golgi-to-plasma membrane trafficking signal, and an ER export motif, resulted in almost 3-fold larger photocurrents (see data for β1-ChR and β1-ChR-YFP) [36]. Consequently, we incorporated a variant of the green fluorescent protein (GFP or YFP), and in most constructs, at least one of both regulatory motifs thought to enhance trafficking and plasma membrane expression (Figure 1; for detailed description and for the exact amino acid composition of the six constructs, see “Appendix A”).

**Figure 1 membranes-12-00907-f001:**
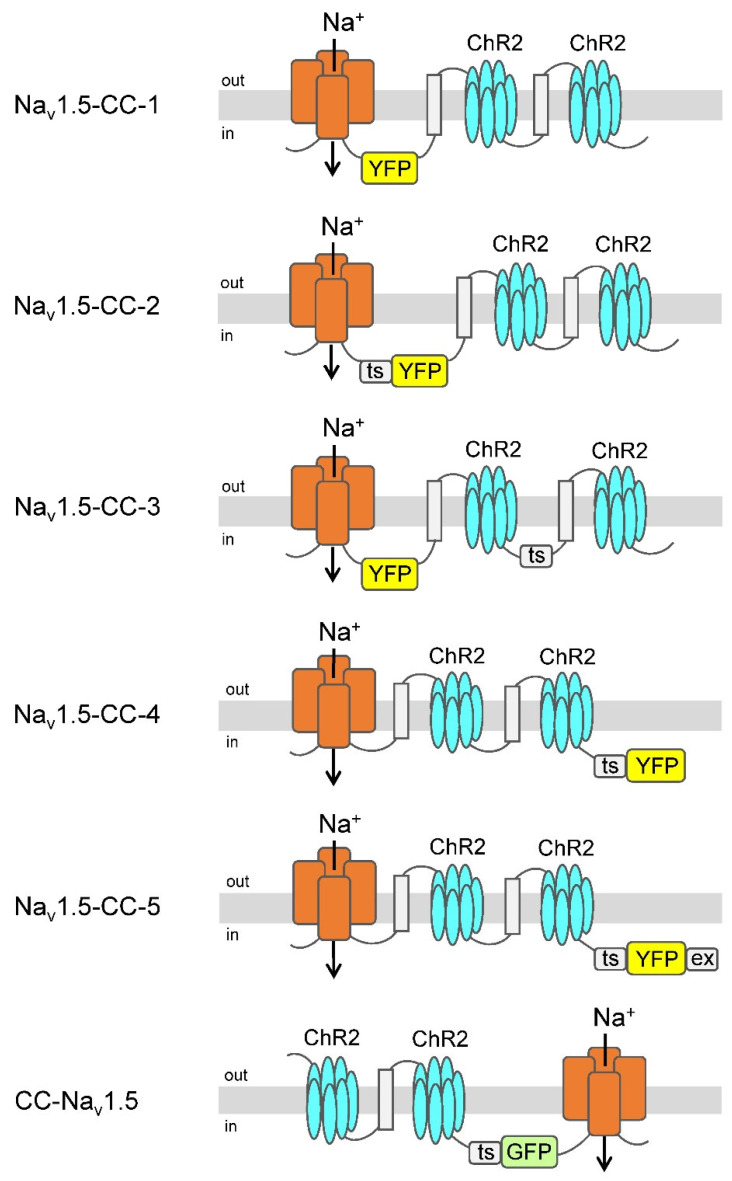
Light-sensitive voltage-gated Na^+^ channels. ChR2 was coupled as a dimer either to the C- or N-terminus of the human cardiac voltage-gated Na^+^ channel (Na_v_1.5). The seven transmembrane regions of ChR2 (blue), the four large domains of Na_v_1.5 (red), and the membrane-spanning region of the β subunit of the human H^+^/K^+^-ATPase (light gray) including the flanking linker peptide [GGGS]_3_ (symbolized by a black line) are illustrated. Variants of the green fluorescent protein (YFP or GFP) were incorporated either into linker regions or at the C-terminal end. Some of the channel constructs contained the Golgi-to-plasma membrane trafficking signal (ts) [48] and, in the case of Na_v_1.5-CC-5, also an ER export motif [49]. For detailed amino acid composition, see “Appendix A”.

Initial expression studies in *Xenopus laevis* oocytes of the six different variants showed robust voltage-activated Na^+^ currents (Figure 2; for current families, see Appendix A). Coupling the ChR2 dimer to the Na_v_1.5 C-terminus (Na_v_1.5-CC-1 to −5), we observed peak current amplitudes larger than 10 µA in all five variants (Table 1), irrespective of whether or not the trafficking signal or the ER export motif was incorporated and independent of the YFP position. Coupling the ChR2 dimer to the Na_v_1.5 N-terminus, however, resulted in a significant Na^+^ current reduction (CC-Na_v_1.5 in Figure 2 and Table 1). To test for functionality of the attached ChR2 dimer, we applied a blue-light pulse from a single LED. We observed the typical ChR2 inward currents in all six variants (Figure 2). Similarly, as noticed already for the voltage-activated Na^+^ current, the smallest photocurrents were detected in CC-Na_v_1.5 (Table 1). The ratios of the voltage-activated peak Na^+^ current (*I_Na_*) and the transient ChR2-mediated photocurrent (*I_ChR2_*) were indistinguishable among the C-terminal fusion channels (Na_v_1.5-CC-1 to −5 in Table 1). In CC-Na_v_1.5, however, this ratio was significantly increased, suggesting that the ChR2 function was impaired in this construct to a greater degree than the Na^+^ channel function.

**Table 1 membranes-12-00907-t001:** Peak current amplitudes, *I_Na_*: *I_ChR2_* ratios, and number of measurements in *Xenopus laevis* oocytes. The amount of cRNA injected per oocyte was about 5 ng. In these initial screening experiments, we used a single LED and a light pulse duration of 1 s.

Channel Construct	Na^+^ Current *I_Na_* (µA)	Photocurrent *I_ChR2_* (nA)	Ratio *I_Na_*: *I_ChR2_*	Number of Oocyte Batches	Number of Measurements
Na_v_1.5-CC-1	13.1 ± 0.8	112.6 ± 20.5	167.5 ± 28.8	3	13
Na_v_1.5-CC-2	12.4 ± 1.1	147.1 ± 26.8	124.2 ± 19.7	3	15
Na_v_1.5-CC-3	11.2 ± 1.0	135.9 ± 17.5	99.9 ± 13.0	3	15
Na_v_1.5-CC-4	12.8 ± 1.5	112.4 ± 14.2	121.8 ± 9.9	3	13
Na_v_1.5-CC-5	12.7 ± 1.3	130.7 ± 17.1	107.7 ± 11.3	3	13
CC-Na_v_1.5	5.6 ± 0.5 *	22.0 ± 3.4 *	485.5 ± 90.1 **	3	31

* significantly smaller than for all other variants (*p* < 0.05); ** significantly larger than for all other variants (*p* < 0.05).

**Figure 2 membranes-12-00907-f002:**
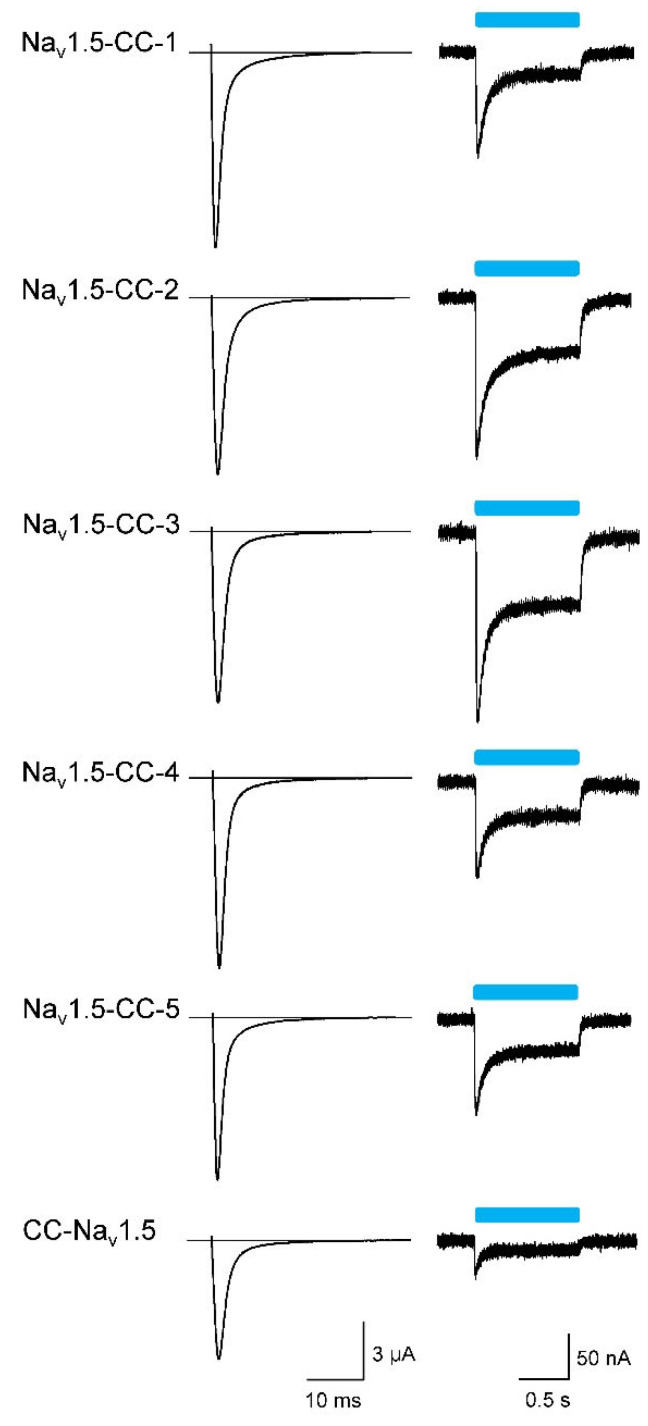
Representative Na^+^ currents and light-activated photocurrents of the six fusion channels. Whole-cell currents were recorded three days after injecting about 5 ng cRNA per *Xenopus* oocyte at a test potential of −20 mV (left; holding potential −100 mV) and using a 1-s blue-light pulse (right; holding potential −80 mV). For current data, number of measurements, and statistics, see Table 1.

As the next step, we addressed the question of whether or not the Na_v_1.5 kinetics were affected in the fusion channels. Because we could not find significant differences in the peak currents between the C-terminal fusion channels Na_v_1.5-CC-1 to −5, we focused on two of them and on CC-Na_v_1.5. To minimize voltage errors in oocytes with a very high Na_v_1.5 expression, we diluted the cRNA preparations before injection, so that the whole-cell Na^+^ currents were smaller than 5 µA. We found that the Na^+^ channel kinetics in the fusion channels were nearly identical to those of wild-type Na_v_1.5 (Table 2; Appendix A”). In Na_v_1.5-CC-2, steady-state activation and inactivation as well as recovery from inactivation and inactivation kinetics were indistinguishable from the wild-type Na_v_1.5 (Table 2). In Na_v_1.5-CC-5 and CC-Na_v_1.5, most of the kinetic parameters remained unchanged, except for a reduction of the slow recovery time constant in Na_v_1.5-CC-5 and the smaller slope values of steady-state activation and inactivation in CC-Na_v_1.5 (Table 2).

### 3.2. Light-Induced Action Potentials

Next, we tried to elicit the action potentials (APs) in oocytes expressing Na_v_1.5-CC-2 or Na_v_1.5-CC-5 by 50 ms blue-light pulses. To illuminate a larger oocyte surface and thus to increase the photocurrent and the likelihood to reach the threshold potential, we used an additional blue LED. Depending on oocyte quality and expression level, whole-cell recordings in the current-clamp mode produced either subthreshold depolarization or typical light-induced APs (Figure 3). Using Na_v_1.5-CC-2 and Na_v_1.5-CC-5, we observed APs in 68 out of 131 oocytes (51.9%) and in 14 out of 78 oocytes (17.9%), respectively. Statistical analysis showed that both fusion channels generated similar APs with respect to threshold potential, upstroke velocity, overshoot potential, and AP duration (Table 3). In CC-Na_v_1.5, however, we tested 97 oocytes from six different oocyte batches, but we were not able to evoke even a single AP (Figure 3).

**Figure 3 membranes-12-00907-f003:**
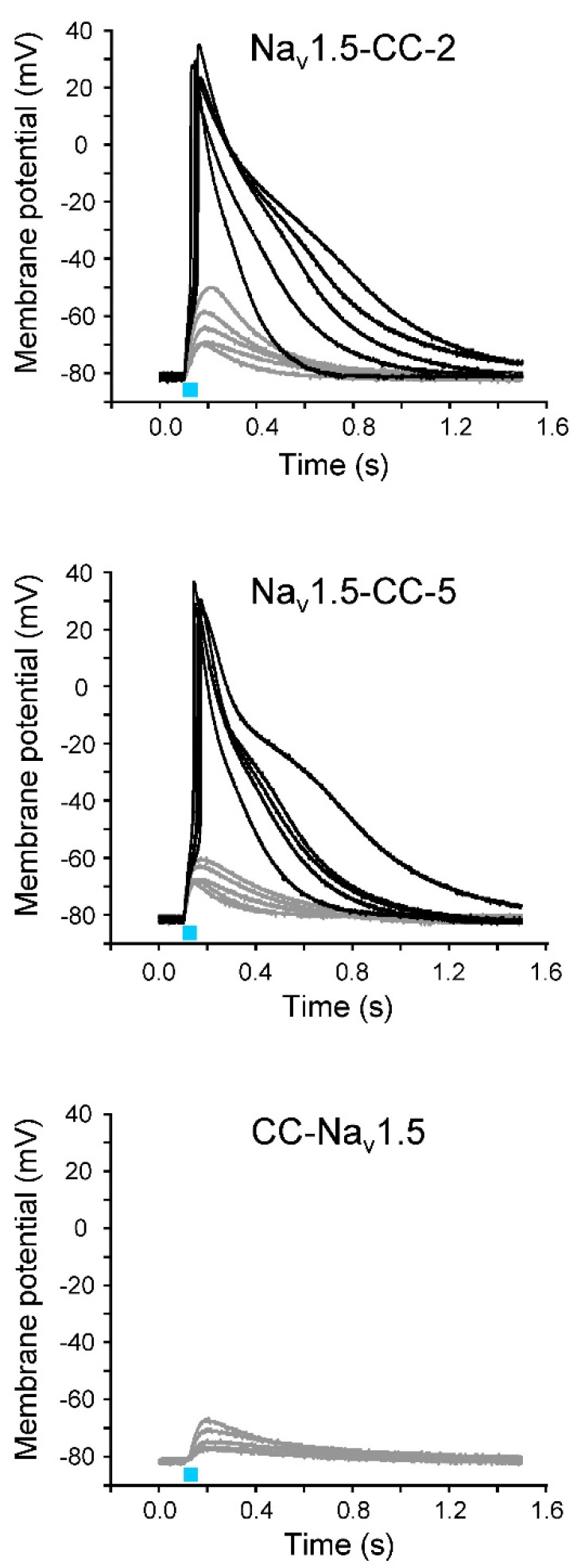
Representative light-induced APs (black) and subthreshold membrane depolarizations (gray) in *Xenopus* oocytes expressing Na_v_1.5-CC-2, Na_v_1.5-CC-5, and CC-Na_v_1.5. The cRNA amount injected per oocyte was about 5 ng. The membrane potential was set to −80 mV, and the change of the membrane voltage was followed in the current-clamp mode upon a 50 ms blue-light pulse (blue bar). The light flash was delivered simultaneously to each oocyte from two LEDs. For detailed statistics on AP parameters, see Table 3.

In order to define the *I_Na_*: *I_ChR2_* ratio and to identify a threshold area for triggering an AP, we plotted the voltage-gated peak *I_Na_* against the corresponding *I_ChR2_* for 131 oocytes expressing Na_v_1.5-CC-2 (Figure 4). In a fusion channel, one would expect a constant *I_Na_*: *I_ChR2_* ratio. Consequently, considering the observed variation in fusion channel expression, a direct proportionality between both current components should be observed. As shown in Figure 4, there was indeed a linear correlation. The slope of the regression line was 21.25 ± 1.39, suggesting a nearly 20-fold larger peak Na^+^ current through the Na_v_1.5 domain in Na_v_1.5-CC-2, compared to the light-induced photocurrent. The gray box in Figure 4 illustrates the AP threshold area, defined by the smallest peak *I_Na_* sufficient for AP generation (7 µA), the largest subthreshold *I_Na_* (14 µA), the smallest photocurrent sufficient for AP generation (170 nA), and the largest subthreshold photocurrent (330 nA).

**Figure 4 membranes-12-00907-f004:**
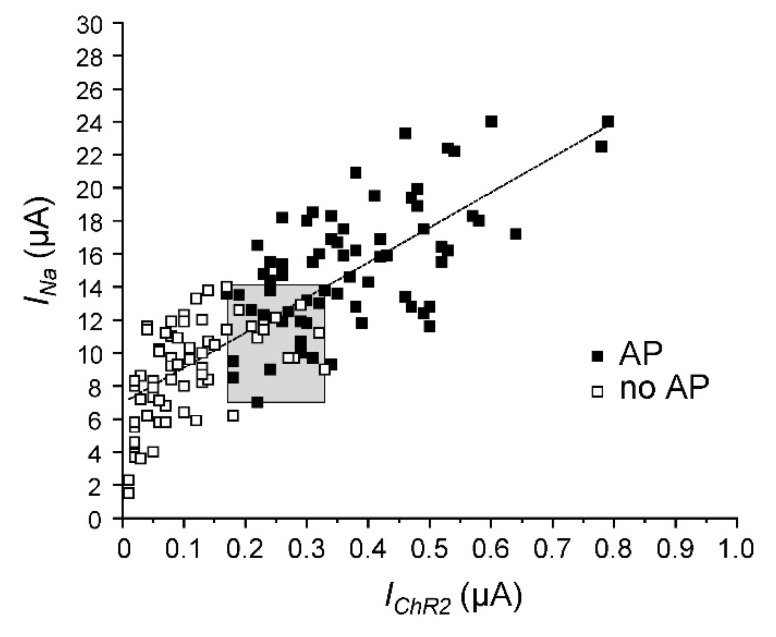
Correlation between peak Na^+^ current (*I_Na_*) and light-induced photocurrent (*I_ChR2_*) obtained for Na_v_1.5-CC-2. Each data point represents both current measurements from a single oocyte. We used a total of 131 oocytes from 6 different batches. Subthreshold depolarizations were obtained in 63 oocytes (open squares). APs were recorded in 68 oocytes (filled squares). Although we tried to keep constant conditions for cRNA injection and current measurement, a considerable variation in channel expression was noticed among these 6 batches of oocytes. Nonetheless, the linear fit indicates that the *I_Na_*: *I_ChR2_* ratio was relatively constant (y = 21.25x + 6.97; R^2^ = 0.63). The gray box illustrates the threshold conditions, defined by the smallest *I_Na_* or *I_ChR2_* that was still sufficient for AP generation, and by the largest *I_Na_* or *I_ChR2_* that produced only subthreshold depolarization. The corresponding range was between 7 to 14 µA for *I_Na_*, and 170 to 330 nA for *I_ChR2_*. The straight line does not run through the zero point of the diagram, indicating an underestimation of *I_ChR2_*. This was most likely due to technical limitations to deliver the blue-light pulse to the whole oocyte surface.

### 3.3. Modulation of AP Shape by Shaker-Related K_v_1.2 Channels

Light-induced APs with Na_v_1.5-CC-2 and Na_v_1.5-CC-5 in oocytes were considerably longer than those naturally occurring in neurons or even in cardiomyocytes. Whereas the upstroke velocity was comparatively fast (see Table 3), we often noticed time intervals for repolarization of more than one second (Figure 3). In our optogenetic oocyte system, repolarization strongly depends on endogenous electrical conductivities. *Xenopus* oocytes obviously lack a sufficient number of respective voltage-activated ion channels that would allow for a fast return to the resting membrane potential. This opens up the possibility to shape the AP, and in particular, to study the effect of individual voltage-gated K^+^ channels on repolarization.

We first tested K_v_1.2, a member of the shaker-related subfamily of delayed rectifier K^+^ channels that is responsible for efficient repolarization in neurons (Figure 5) [37]. K_v_1.2 generated rapidly activating outward currents (Figure 5B). When co-expressing Na_v_1.5-CC-2 with K_v_1.2, we found substantially accelerated repolarization of light-triggered APs in the current-clamp mode, when compared to oocytes expressing Na_v_1.5-CC-2 alone (Figure 5A). Membrane potentials changed rapidly from the overshoot potential of nearly +20 mV to −50 mV within less than 20 ms. This effect of K_v_1.2 was observed in all cells investigated (*n* = 15; 5 batches of oocytes). The AP duration (APD) was significantly shortened at -10 mV and −40 mV by nearly 95%, and at −70 mV by nearly 50% (for APD values see Table 3). As shown in Figure 5C, the effect on APD depended on the K_v_1.2 expression level: The larger the outward K^+^ current, the shorter the APD. In addition to the effect on the repolarization phase, the fast-activating K_v_1.2 outward currents also modulated the AP upstroke. We found a reduced AP upstroke velocity and a less positive overshoot potential, compared to APs generated in the absence of the delayed rectifier (Table 3). The threshold potential, however, remained unchanged (Table 3).

**Figure 5 membranes-12-00907-f005:**
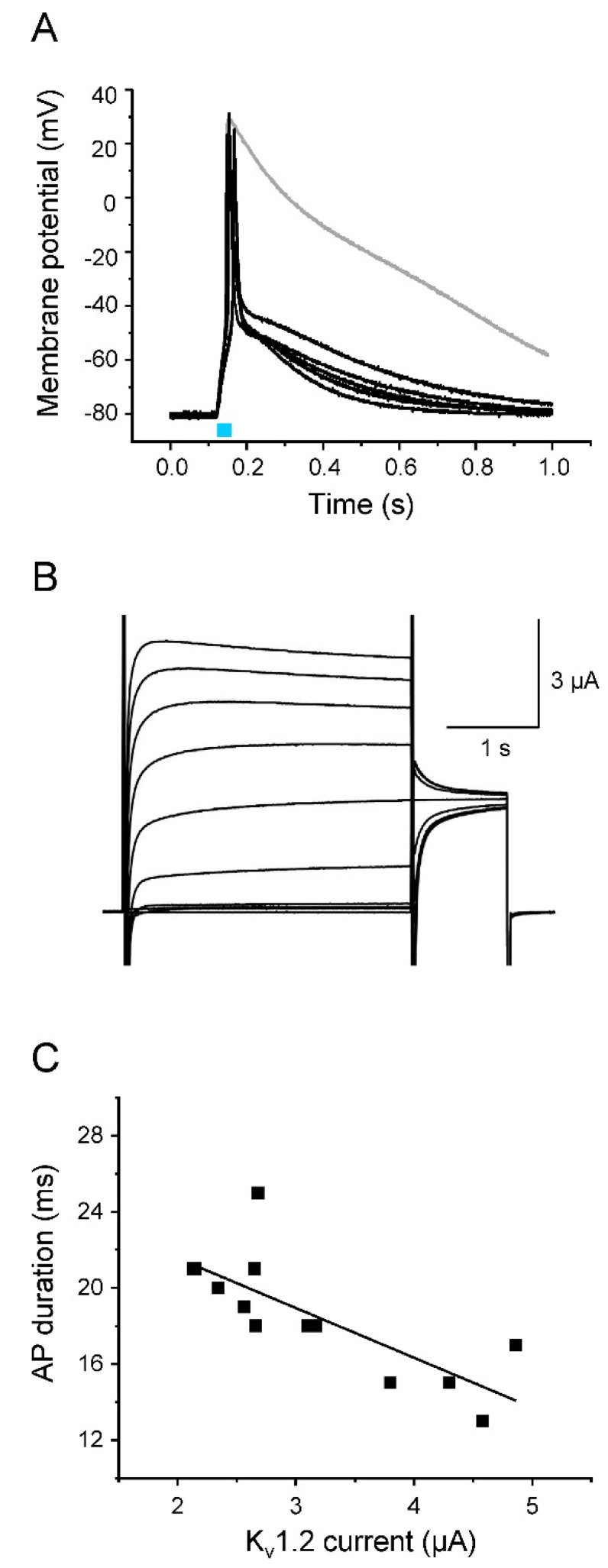
Acceleration of AP repolarization by K_v_1.2. (**A**) Representative light-induced APs from 5 different oocytes co-expressing Na_v_1.5-CC-2 and K_v_1.2. The 30 ms-light pulse is illustrated as a blue bar. The gray line represents a control AP from an oocyte injected with Na_v_1.5-CC-2 cRNA only. For detailed statistics on AP parameters, see Table 3. Peak current values and oocyte features are summarized in “Appendix A). (**B**) Corresponding K_v_1.2 currents elicited by a first test pulse between −50 mV and +40 mV in 10 mV increments, followed by a second test pulse to 0 mV (holding potential −100 mV). (**C**) Correlation between AP duration, measured in the current-clamp mode at −40 mV, and the corresponding K_v_1.2 plateau current, recorded in the voltage-clamp mode at +10 mV (13 oocytes from 5 different oocyte batches). The line represents the results of a linear fit (y = −2.61x + 26.77, R^2^ = 0.59).

Whereas the initial repolarization was significantly enhanced by the co-expression of K_v_1.2, the following return of the membrane potential from about −50 mV to the resting potential of −80 mV was similarly slow as in the absence of K_v_1.2 (compare Figure 3 and Figure 5A). This second repolarization phase obviously depended on endogenous channels, even in the presence of K_v_1.2. As shown in Figure 5B, K_v_1.2 channels do not significantly inactivate during the test pulses, implicating a pronounced repolarizing outward current also at a later phase of repolarization. However, we found in parallel experiments a mid-activation potential of −9.88 ± 1.44 mV (*n* = 6). Consequently, K_v_1.2 channels should be largely closed at potentials more negative than −40 mV, so that they cannot promote repolarization to the resting potential. To prove this assumption, we measured the K^+^ current at -50 mV after short depolarized test potentials typically occurring during the AP upstroke and overshoot. In Figure 6, we show a representative light-induced AP (Figure 6A), and plots in the same timescale of the corresponding photocurrent (Figure 6B) as well as the ionic currents elicited by short test pulses to −20 mV, 0 mV, and +20 mV, followed by a second pulse to −50 mV (Figure 6C). The photocurrent generated by a 30 ms blue-light pulse caused an initial membrane depolarization that was followed by a fast AP upstroke (compare Figure 6A,B). Voltage-clamp recordings in the same oocyte revealed a corresponding early inward current through Na_v_1.5-CC-2 channels, when stepping from −80 mV to the three test potentials (open triangle in Figure 6C). This large inward current and the fast current decay were due to the activation and inactivation of the Na_v_1.5 domain in Na_v_1.5-CC-2. Shortly thereafter, K_v_1.2 was activated producing a large outward current. Stepping back to −50 mV immediately after the initial 15 ms pulses completely suppressed this rectifying current (arrow in Figure 6C). Consequently, K_v_1.2 channels are closed at a later repolarization phase, and only the endogenous channels were responsible for the slow return to the resting membrane potential (Figure 5A and Figure 6A).

**Figure 6 membranes-12-00907-f006:**
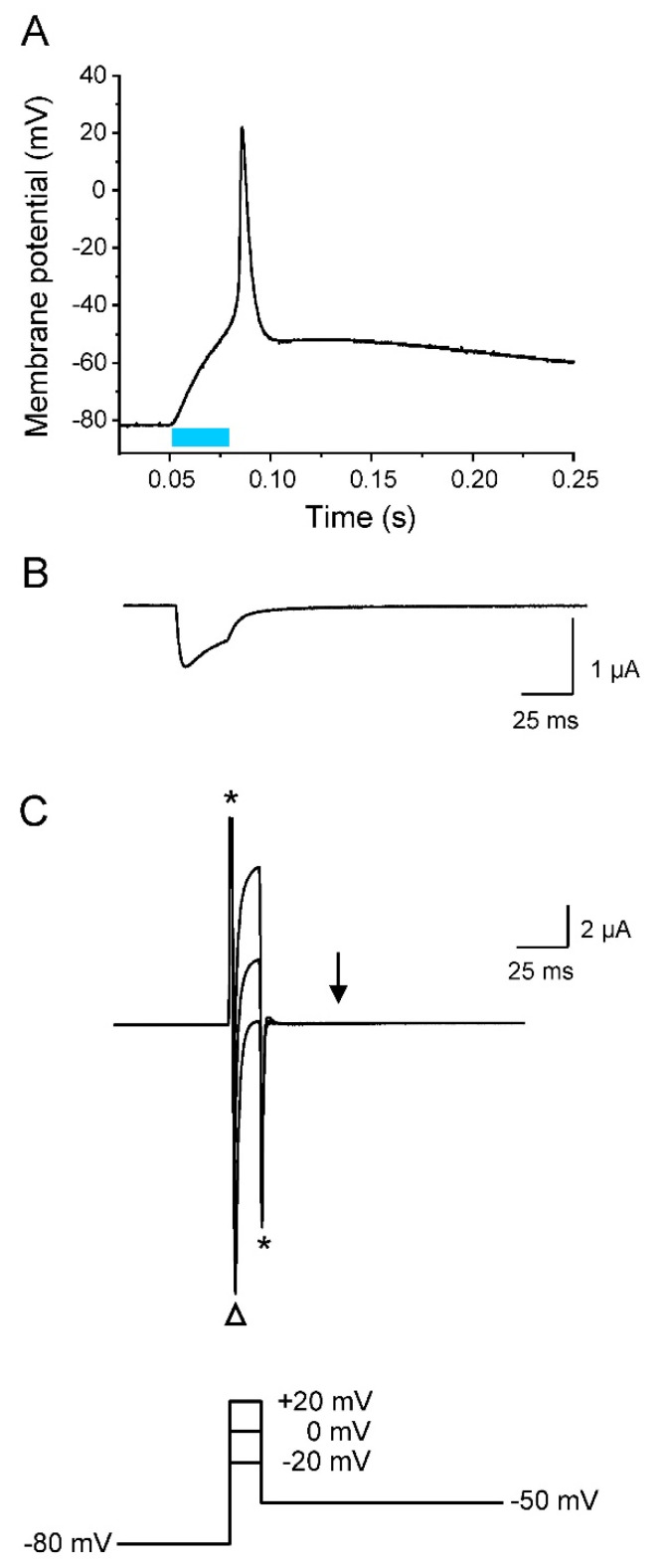
Early but not late AP repolarization by K_v_1.2. Recordings were performed on the same oocyte co-expressing Na_v_1.5-CC-2 and K_v_1.2, and plotted using the same timescale in (**A**–**C**). (**A**) Current-clamp recording (light pulse duration 30 ms). (**B**) Corresponding whole-cell photocurrent recorded in the voltage-clamp mode. This light-induced current component caused the initial membrane depolarization seen in A. (**C**) Whole-cell ionic currents elicited by three different 15 ms depolarized pulses followed by a second pulse to −50 mV. Asterisks mark the capacitive currents. The open triangle indicates the fast Na^+^ inward current through Na_v_1.5-CC-2, responsible for AP upstroke. Test pulses to 0 and +20 mV caused large outward currents through K_v_1.2 channels that are responsible for the fast initial repolarization phase to −50 mV seen in A. Stepping back to −50 mV, however, completely abolished any rectifying outward current (arrow).

### 3.4. Modulation of AP Shape by hERG Channels

Next, we co-expressed Na_v_1.5-CC-2 and hERG, a potassium channel that is responsible for the rapid delayed rectifier current *I*_Kr_ in the heart (Figure 7). Similarly as seen for K_v_1.2 co-expression, we found substantially accelerated repolarization of all light-triggered APs in the current-clamp mode. APDs were significantly shorter at −10 mV, −40 mV, and −70 mV, when compared to values found for control oocytes injected with Na_v_1.5-CC-2 cRNA only. Again, we noticed a reduced upstroke velocity, a less positive overshoot potential, and an unchanged threshold potential, compared to APs generated in the absence of hERG (Table 3).

**Figure 7 membranes-12-00907-f007:**
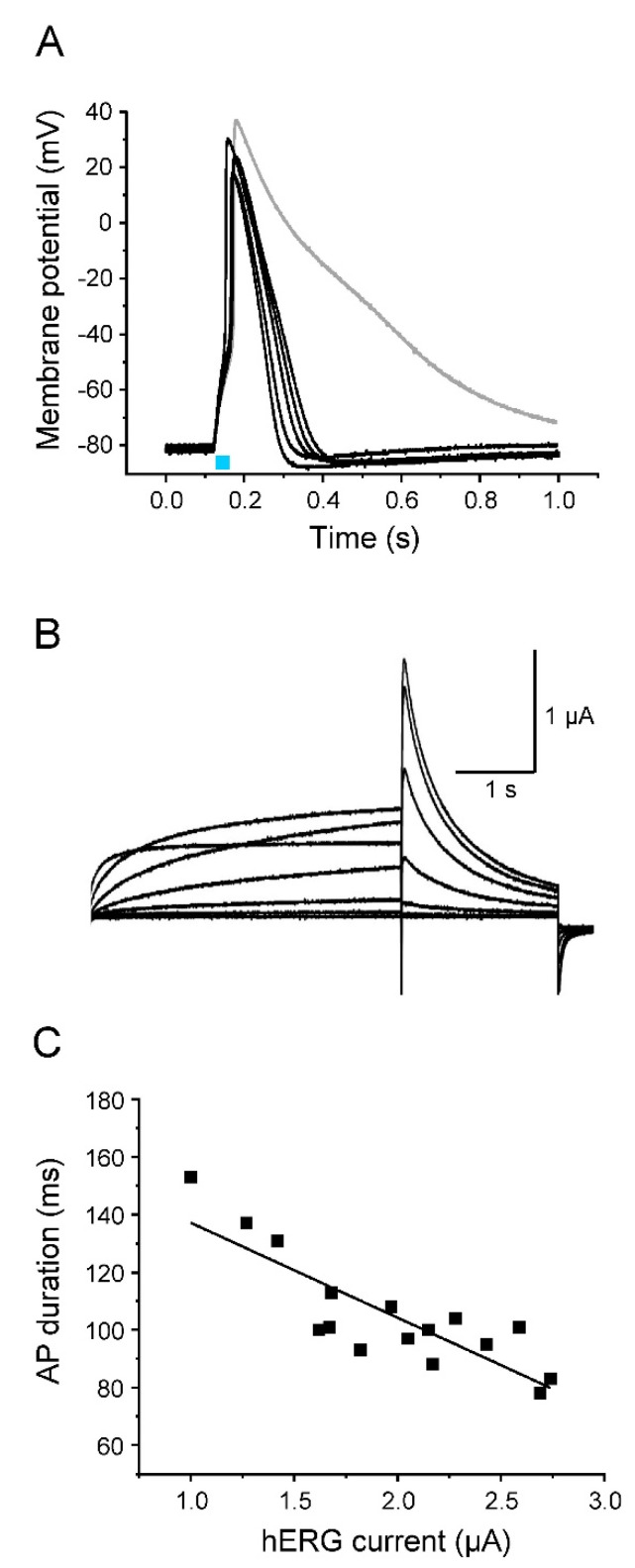
Acceleration of AP repolarization by hERG. (**A**) Representative light-induced APs from 5 different oocytes co-expressing Na_v_1.5-CC-2 and hERG (light pulse 30 ms). The gray line represents a control AP from an oocyte injected with Na_v_1.5-CC-2 cRNA only. For detailed statistics on AP parameters, peak current data, and oocyte features, see Table 3 and “Appendix A. (**B**) Corresponding hERG currents elicited by prepulses between −50 mV and +10 mV in 10 mV increments, followed by a second test pulse to −50 mV (holding potential −90 mV). (**C**) Correlation between AP duration, measured in the current-clamp mode at −40 mV, and the corresponding hERG peak current at −50 mV, recorded in the voltage-clamp mode (prepulse: +10 mV for 4 s; *n* = 16 from 5 different batches of oocytes). The line represents the results of a linear fit (y = −32.92x + 170.05, R^2^ = 0.71).

In contrast to APs generated with K_v_1.2, however, initial repolarization was less efficient (see APD values at −10 mV and −40 mV in Table 3). At the same time, we found complete repolarization to the resting membrane potential, and in most oocytes, even a noticeable hyperpolarization. APD values at −70 mV were significantly shorter, compared to oocytes co-expressing K_v_1.2 (Table 3). In contrast to other K_v_ channels including K_v_1.2, hERG channels display unique electrophysiological properties [50,51]. Membrane depolarization causes slow activation, but very rapid inactivation. Consequently, initial test pulses up to +40 mV produced only moderate outward currents (Figure 7B). Membrane repolarization, however, causes fast recovery from the inactivated state, but very slow channel deactivation, allowing hERG channels to remain in the conductive state for a prolonged time. Consequently, stepping back from positive voltages to -50 mV produced larger outward currents (Figure 7B). The size of the respective peak current could be again inversely correlated to the AP duration (Figure 7C).

To prove that the unusual gating properties of hERG are responsible for efficient repolarization, we measured the K^+^ current at -50 mV after short depolarized test potentials typically occurring during AP upstroke and overshoot (Figure 8). Similarly as for Figure 6, we recorded in the same oocyte a light-induced AP (Figure 8A), the corresponding photocurrent (Figure 8B) as well as the ionic currents, elicited by the indicated pulse protocol (Figure 8C). As in case of K_v_1.2 co-expression, we found a large light-activated photocurrent (Figure 8B), as well as an inward current through the Na_v_1.5 domain, when stepping from −80 mV to the three test potentials (open triangle in Figure 8C). During these voltage steps up to +20 mV, a delayed outward current component was detectable. Compared to oocytes co-expressing K_v_1.2 (Figure 6B), this current was smaller, finally resulting in a less efficient early repolarization. Because hERG channels undergo a fast transition from the inactivated to a relatively stable open state during AP repolarization, stepping back to −50 mV from +20 mV resulted in increased outward currents (arrow in Figure 8C). Consequently, hERG channels are open at a later repolarization phase, allowing for the complete return to the resting membrane potential.

**Figure 8 membranes-12-00907-f008:**
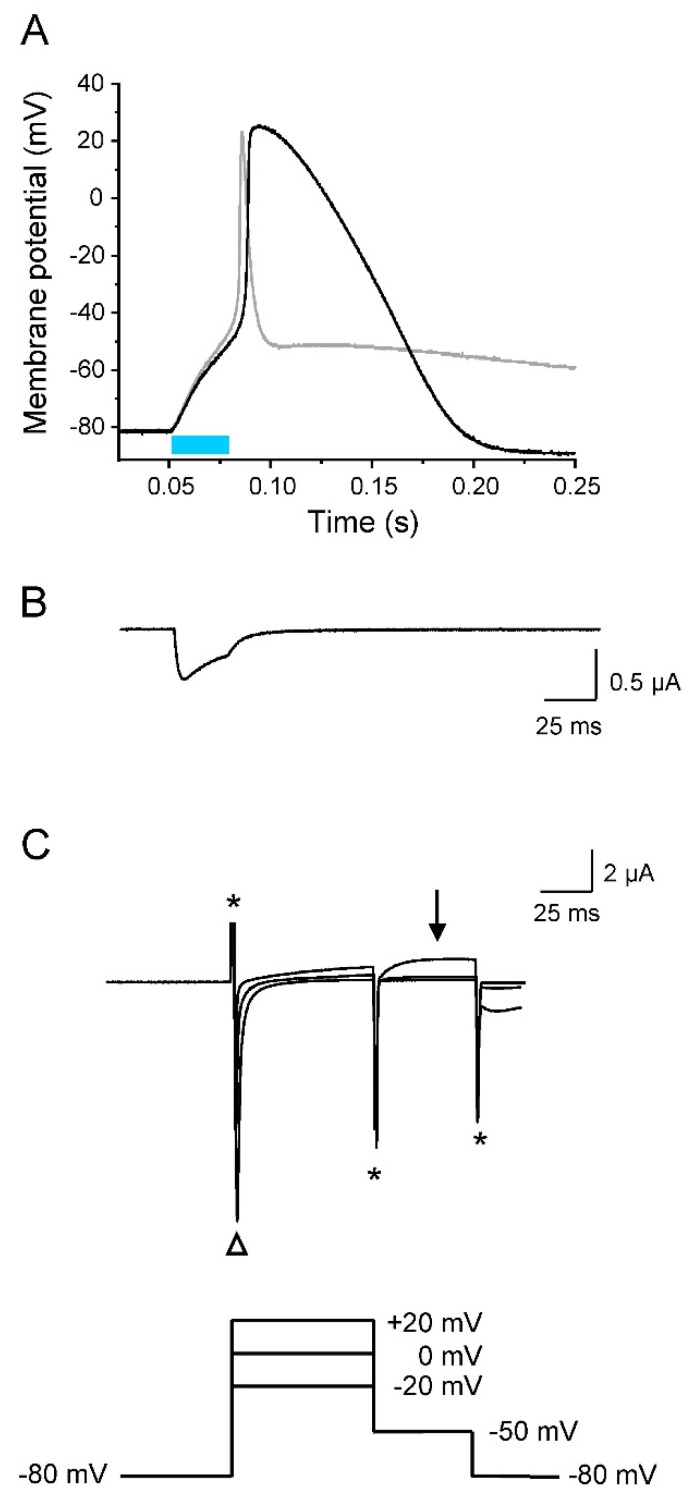
Pronounced late AP repolarization by hERG. All recordings were performed on the same oocyte co-expressing Na_v_1.5-CC-2 and hERG. Plots are in the same timescale. (**A**) Current-clamp recording (light pulse duration 30 ms). For comparison, the gray line represents an AP obtained after co-expression of K_v_1.2 (shown in Figure 6A). (**B**) Corresponding whole-cell photocurrent (voltage-clamp mode), resulting in initial membrane depolarization to the threshold potential. (**C**) Whole-cell ionic currents elicited by three different prepulses followed by a second pulse to −50 mV. Asterisks indicate the capacitive currents. The open triangle indicates the fast Na^+^ inward current through Na_v_1.5-CC-2, responsible for AP upstroke. In contrast to K_v_1.2, initial prepulses caused small outward currents through hERG channels. Correspondingly, early AP repolarization is less efficient (see also data for APD −10 mV and −40 mV in Table 3). Stepping back to −50 mV from +20 mV, however, resulted in hERG activation, pronounced outward rectification (arrow), and hyperpolarization of the oocyte membrane (seen in (**A**) and in Figure 7A).

### 3.5. Modulation of AP Shape by LQT3 Deletion Variant ΔKPQ

After analyzing the effect of two different K_v_ channels on light-induced APs, we addressed the question of whether or not AP prolongation is detectable using LQT3 mutant channels. For this purpose, we introduced the ΔKPQ deletion in the Na_v_1.5-CC-2 background. *Xenopus* oocytes, injected with the resulting Na_v_1.5-CC-ΔKPQ cRNA, produced considerably longer APs compared to control oocytes expressing Na_v_1.5-CC-2 (Figure 9A). APD at −10 mV and −40 mV increased nearly 5- to 10-fold (Table 3). We simultaneously recorded the non-inactivating Na^+^ current fraction in Na_v_1.5-CC-ΔKPQ channels (Figure 9B). We found a direct proportionality between APD at −40 mV and the size of this persistent Na^+^ current component (Figure 9C). In addition to prolonged APs, we also recorded a positive shift of the threshold potential, a reduced AP upstroke velocity, and a less positive overshoot potential, compared to control APs generated after expression of Na_v_1.5-CC-2 or Na_v_1.5-CC-5 (Table 3).

**Figure 9 membranes-12-00907-f009:**
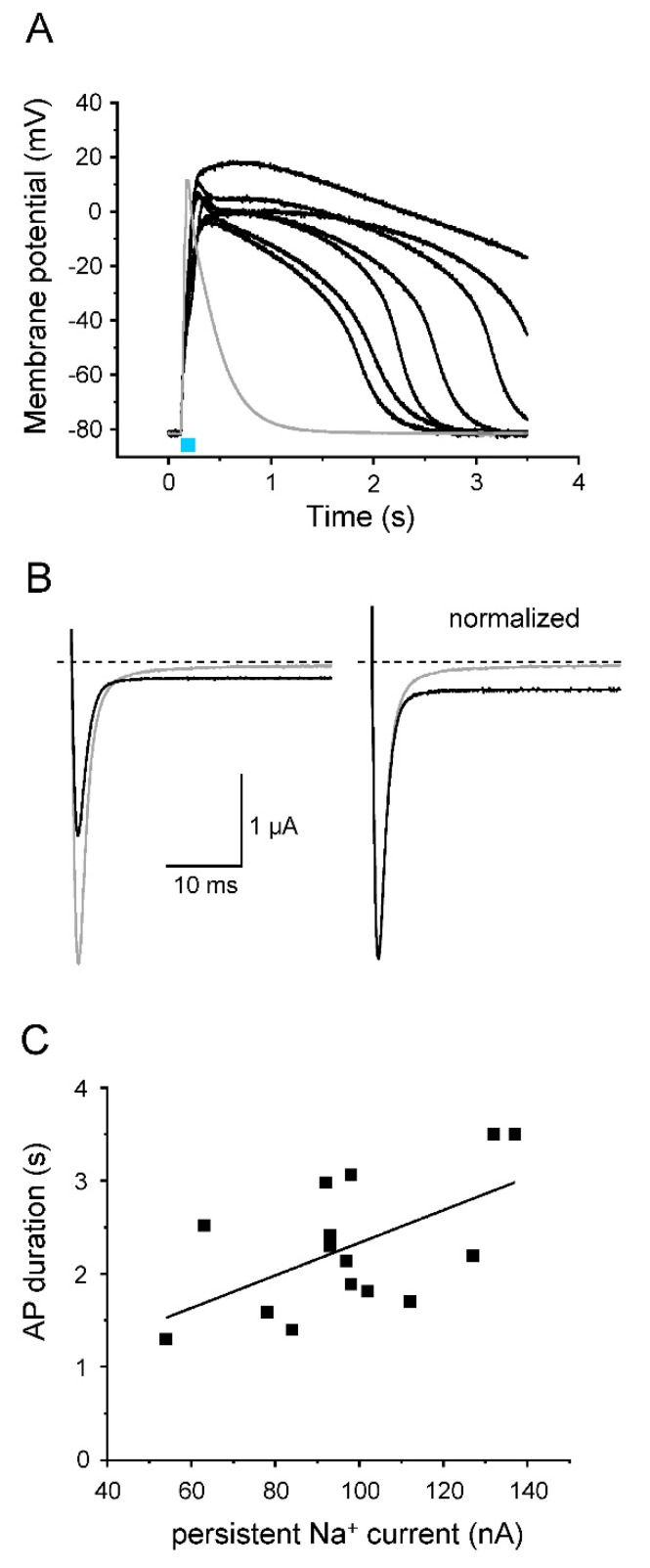
Prolongation of AP repolarization by LQT3 deletion variant ΔKPQ. (**A**) Representative light-induced APs from 7 different oocytes expressing Na_v_1.5-CC-ΔKPQ. The 30 ms-light pulse is illustrated as a blue bar. The gray line represents an AP from a control oocyte injected with Na_v_1.5-CC-2. For detailed statistics, see Table 3 and “Appendix A). (**B**) Corresponding whole-cell Na^+^ currents elicited by a test pulse to −20 mV (holding potential −120 mV; gray line: Na_v_1.5-CC-2; black line: Na_v_1.5-CC-ΔKPQ; right: current traces normalized to the peak current). In Na_v_1.5-CC-ΔKPQ, the non-inactivating current fraction was significantly increased (up to 7% of the transient current). At the same time, we noticed a reduced peak current compared to Na_v_1.5-CC-2. (**C**) Correlation between AP duration, measured in the current-clamp mode at −40 mV, and the corresponding persistent current at −20 mV, recorded in the voltage-clamp mode (*n* = 15 from 6 different batches of oocytes). The line represents the results of a linear fit (y = 0.02x; R^2^ = 0.94).

## 4. Discussion

The great potential of optogenetics is the non-invasive induction of AP firing in naturally excitable or non-excitable cells [27,30,52,53]. In the present study, we provide novel optical switches for this growing research area by combining the light-sensitive feature of ChR2 with the high ion conductivity of a voltage-gated Na^+^ channel. Using fusion proteins, we were able to elicit APs in normally non-excitable *Xenopus* oocytes by short blue-light flashes. Together with our previously reported β1-ChR2 constructs [36], we have now two distinct tools for directing ChR2 in close proximity to Na^+^ channels to elicit APs. As shown previously by several other groups [24,29,30,31,32,33,43], supra-threshold membrane depolarization can also be achieved by expressing ChR2 alone in heterologous host cells or by co-expressing ChR2 with endogenous Na^+^ channels of naturally excitable cells. The Na_v_1.5-CC fusion constructs reported in this study are beneficial, in that a single construct is sufficient. Voltage- and light-activated currents are two major determinants for threshold potential and AP upstroke. Consequently, more standardized APs can be obtained when the ratio of *I_Na_*: *I_ChR2_* is relatively constant. By studying the modulatory impact of two distinct K^+^ channels and one of the best-characterized LQT3 mutation on the AP shape, we show an application possibility for our new optogenetic tools.

When Na_v_1.5-CC-2 is co-expressed with either K_v_1.2 or hERG, characteristic AP alterations were noticed. Whereas the threshold potential remained unchanged, we first found a reduced upstroke velocity and less positive overshoot potentials. For this early K^+^ channel effect, three parameters are relevant: (1) the density of K^+^ channels in the plasma membrane, which was similar for both channel types (Appendix A), (2) the voltage range of steady-state activation, which can be estimated from the half-maximal activation voltage V_1/2_, and (3) the activation kinetics, which can be characterized by the rise time during activation [37], by fitting the activation time course [54], or by the time required to reach the half-maximal current after a certain test pulse was started (t_1/2_) [55]. In both K_v_1.2 and hERG, V_1/2_ values for oocyte expression are equal or more negative than -10 mV. At the same time, channel activation occurs within a few milliseconds, in particular when the membrane approaches the overshoot potential [37,54,55]. Consequently, the AP rise time of about 1 ms could be indeed long enough to induce a small but significant K^+^ outward current, counteracting the massive influx of Na^+^ and thereby decelerating the upstroke velocity and reducing the AP amplitude. The most notable effect of K^+^ channel co-expression was unequivocally the marked acceleration of AP repolarization, which is the actual physiological function of delayed rectifier K^+^ channels. K_v_1.2 and hERG channels showed characteristic effects, due to their distinct channel kinetics. K_v_1.2 currents were highly efficient in shortening the early repolarization phase because the channels rapidly open in a voltage-dependent manner and show little or no inactivation. Consequently, initial repolarization from the overshoot potential to about −50 mV occurred within a few milliseconds. However, at membrane potentials more negative than −50 mV, currents through K_v_1.2 were absent or too small to account for further acceleration of repolarization (see Figure 5 and Figure 6). In contrast, hERG channels generate small initial outward currents upon depolarization, because they rapidly inactivate from the open state [14,50,51]. Consequently, initial repolarization is less pronounced. However, the more negative the membrane potential, the more the hERG channels return from the inactivated to the open state, before slowly returning to their closed conformation. The resulting K^+^ flux through hERG channels is essential for restoring the resting membrane potential, as seen in Figure 7 and Figure 8. In case of both K_v_1.2 and hERG co-expression, AP duration inversely correlated with the size of the K^+^ current. This observation confirms the essential nature of delayed rectifiers for AP length, and moreover, even resembles the situation in patients carrying loss-of-function mutations in hERG and diagnosed with LQT2 [56,57,58].

In addition to the specific effects of the two individual K^+^ fluxes on AP shape, we also tested whether or not our oocyte model is sensitive enough to detect the well-established effect of a persistent Na^+^ current on AP duration [18,22,59,60,61]. We indeed noticed a correlation between the size of the non-inactivating Na^+^ current fraction and the AP duration for Na_v_1.5-CC-ΔKPQ (see Figure 9). AP prolongation was markedly pronounced, suggesting that our novel oocyte system could be sensitive in detecting the effect of much milder inactivation defects. In future experiments, it could be challenging to introduce various other Na^+^ channel mutations, associated with life-threatening arrhythmias, and to test the effect of antiarrhythmic drugs such as mexiletine or ranolazine [61,62,63,64].

There are also a few limitations of our optogenetic *Xenopus* model that should be mentioned. First, the size of the photocurrent and the efficacy to elicit APs depended not only on the oocyte quality and the expression level but also on the surface area exposed to blue light. Using our current setup, it was possible to determine whole-cell peak Na^+^ currents, but it was difficult to illuminate the whole oocyte surface. Consequently, the *I_Na_*: *I_ChR2_* ratios are most likely overestimated, especially when using only a single blue LED (initial screening experiments and values in Table 1). Second, from our experiments, we cannot conclude whether or not only one of the ChR2 channel domains is functional. Previously, we could not obtain functional fusions by coupling a single ChR2 to Na_v_1.5 [36]. In our present study, we successfully used a ChR2 tandem, implicating that only the second ChR2 is functional. However, it is also possible that the first ChR2 domain was stabilized in the dimer and contributed to the observed photocurrent. Third, we still have to adjust the resting potential by current injection and to record the light-induced voltage changes by two invasive microelectrodes. Future co-expression of inward rectifier or tandem pore domain K^+^ channels in combination with optical dyes sensing the membrane voltage are the next challenging steps toward non-invasive AP triggering and detection.

In conclusion, our new optogenetic Xenopus oocyte model offers a fast and reliable system for the generation of light-induced action potentials. It allows for structure–function relation studies of various wild-type and mutant channels and bears the potential for high-throughput drug testing. Ongoing research is aimed at coupling other Na^+^ channel isoforms to ChR2 and at transferring our novel optical switches to mammalian cell lines, stem cells, and cardiomyocytes.

## Figures and Tables

**Table 2 membranes-12-00907-t002:** Electrophysiological properties of three selected Na_v_1.5-ChR2 channels compared to wild-type Na_v_1.5 in *Xenopus* oocytes. The amount of cRNA per oocyte was as follows: 0.2 ng for Na_v_1.5, 0.4 ng for Na_v_1.5-CC-2 and Na_v_1.5-CC-5, and 1 ng for CC-Na_v_1.5. Under these conditions, peak whole-cell Na^+^ currents at −20 mV were as follows: 2.3 ± 0.1 µA for Na_v_1.5, 2.1 ± 0.1 for Na_v_1.5-CC-2, 1.9 ± 0.2 µA for Na_v_1.5-CC-5, and 1.0 ± 0.1 µA for CC-Na_v_1.5. Representative whole-cell Na^+^ current families are shown in “Appendix A. For equations and abbreviations, see Materials and Methods.

Channel	Steady-State Activation		Steady-State Inactivation		Recovery from Inactivation			
	V_m_ (mV)	s (mV)	V_h_ (mV)	s (mV)	τ_f_ (ms)	A_f_	τ_s_ (ms)	A_s_
Na_v_1.5	−25.28 ± 0.77	2.87 ± 0.12	−51.80 ± 0.62	4.95 ± 0.15	3.23 ± 0.04	93.11 ± 0.44	178.22 ± 19.79	6.89 ± 0.44
Na_v_1.5-CC-2	−25.34 ± 0.75	3.09 ± 0.17	−52.34 ± 0.34	5.00 ± 0.18	3.37 ± 0.06	94.12 ± 0.91	164.92 ± 29.19	5.88 ± 0.91
Na_v_1.5-CC-5	−25.99 ± 0.46	3.05 ± 0.28	−52.20 ± 0.74	5.04 ± 0.25	3.21 ± 0.35	94.56 ± 1.05	58.28 ± 8.59 *	5.44 ± 1.05
CC-Na_v_1.5	−24.93 ± 0.78	2.68 ± 0.05 *	−50.76 ± 0.77	4.54 ± 0.06 *	3.21 ± 0.06	90.19 ± 1.31	216.15 ± 29.67	9.81 ± 1.31

* indicates *p* < 0.05 vs. Na_v_1.5 (with *n* = 7–9).

**Table 3 membranes-12-00907-t003:** Statistics on AP parameters. For current amplitudes, oocyte resting membrane potential, and oocyte membrane resistance, see “Appendix A). Number of measurements were at least 12 using at least 4 batches of oocytes. AP upstroke was characterized by the 25–75% rise time and the voltage alteration within this time period.

Channel	Threshold	AP Upstroke		Overshoot	AP Duration at		
Constructs	Potential	Rise Time	Velocity	Potential	−10 mV	−40 mV	−70 mV
	(mV)	(ms)	(mV/ms)	(mV)	(ms)	(ms)	(ms)
Na_v_1.5-CC-2	−47.42 ± 0.45	1.02 ± 0.08	39.36 ± 3.26	27.42 ± 1.99	156.3 ± 15.7	423.6 ± 38.6	832.9 ± 67.3
Na_v_1.5-CC-5	−50.83 ± 0.44	1.05 ± 0.08	40.73 ± 3.93	28.17 ± 1.23	137.5 ± 12.4	404.2 ± 54.7	821.2 ± 75.9
Na_v_1.5-CC-2+K_v_1.2	−48.00 ± 0.40	1.16 ± 0.08	30.83 ± 2.51 *	18.47 ± 1.74 *	8.1 ± 0.61 *	21.4 ± 2.27 *	407.3 ± 20.2 *
Na_v_1.5-CC-2 +hERG	−46.44 ± 0.49	1.40 ± 0.12 *	26.54 ± 2.54 *	19.31 ± 1.62 *	59.6 ± 2.94 *^,^**	105.1 ± 4.99 *^,^**	182.2 ± 10.7 *^,^**
Na_v_1.5-CC-ΔKPQ	−39.28 ± 1.48 *	53.06 ± 5.11 *	0.51 ± 0.06 *	5.56 ± 2.31 *	1420.6 ± 190.08 *	2314.6 ± 165.18 *	2735.6 ± 140.29 *

* *p* < 0.05 compared to Na_v_1.5-CC-2 and Na_v_1.5-CC-5.; ** *p* < 0.05 compared to Na_v_1.5-CC-2 + K_v_1.2.

## Data Availability

The data presented in this study are available on request from the corresponding author.

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
