# Peer review of "Coupling the Cardiac Voltage-Gated Sodium Channel to Channelrhodopsin-2 Generates Novel Optical Switches for Action Potential Studies"

_membranes, 2022, doi:10.3390/membranes12100907_

Round 1
Reviewer 1 Report
The manuscript is good. But need majr revision with regard to grammatical error. Some latet articles not mentioned. The authors must include latest references. Include https://doi.org/10.1016/j.bioorg.2021.105230. Otherwise the manuscript may be accepted after revision.
Author Response
Dear Sir or Madam, Thank you very much for revising our manuscript and for giving us suggestions to improve it. We included some English changes using an online tool. Additionally, we got several helpful comments from a native speaker. As suggested, we also included the recent paper by Pal et al. (2021) in the Discussion part. Yours sincerely, Thomas ZimmerReviewer 2 Report
The working hypothesis of the paper is well-planned, considering the information available from previous studies. Additionally, the adopted methodology is well suited to the research problem addressed by the authors. The results are well-explained and support the authors' discussion and conclusion. Also, the statistical methods used for the analysis of data are appropriate.
One minor comment is that the authors could show the Coefficient of determination in the correlation plot instead of mentioning it in the figure legends. It will help the readers to get a quick inference.
The paper is well suited to this journal and should be accepted in the current format.
Author Response
Dear Sir or Madam, Thank you very much for reading our manuscript and for your favourable review. Yours sincerely, Thomas Zimmer